# Chronic Treatment with Somatostatin Analogues in Recurrent Type 1 Gastric Neuroendocrine Tumors

**DOI:** 10.3390/biomedicines11030872

**Published:** 2023-03-13

**Authors:** Fernando Sebastian-Valles, Blanca Bernaldo Madrid, Carolina Sager, Elena Carrillo López, Sara Mera Carreiro, Laura Ávila Antón, Noelia Sánchez-Maroto García, Miguel Antonio Sampedro-Nuñez, Jose Ángel Díaz Pérez, Mónica Marazuela

**Affiliations:** 1Endocrinology and Nutrition Department, Hospital Universitario de La Princesa, Instituto de Investigación Sanitaria de La Princesa, Universidad Autonoma de Madrid, 28006 Madrid, Spain; 2Endocrinology and Nutrition Department, Hospital Clínico San Carlos and Instituto de Investigación Sanitaria del Hospital Clínico San Carlos (IdISSC), 28040 Madrid, Spain

**Keywords:** gastric neuroendocrine tumors, gastric carcinoids, somatostatin analogs, lanreotide, octreotide

## Abstract

Background: Type 1 gastric neuroendocrine tumors (GC-1) represent an uncommon subtype of neoplasms. Endoscopic resection has been proposed as the treatment of choice; active surveillance may be performed in those smaller than 1 cm, while gastric surgery may be performed for those with frequent recurrences. The antiproliferative effect of somatostatin analogues (SSA) is well known, and their action on GC-1s has been postulated as a chronic treatment to reduce recurrence. Methods: A two-centered, retrospective, observational study that included nine patients (55.6% women) diagnosed with GC-1, receiving long-term treatment with SSA, with a median follow-up from baseline of 22 months, was undertaken. Endoscopic follow-up, extension study, and analytical values of chromogranin A (Cg A) and gastrin were collected. Results: In total, 88.9% of patients presented partial or complete response. Treatment with SSA was the only independent factor with a trend to prevent tumor recurrence (Odds Ratio 0.054; *p* = 0.005). A nonsignificant tendency toward a decrease in CgA and gastrin was observed; lack of significance was probably related to concomitant treatment with proton pump inhibitors in some patients. Conclusions: Chronic treatment with SSA is a feasible option for recurrent GC-1s that are difficult to manage using endoscopy or gastrectomy. Randomized clinical trials to provide more scientific evidence are still needed.

## 1. Introduction

Gastro-entero-pancreatic neuroendocrine tumors (GEP-NETs) constitute a heterogeneous group of neoplasms with low incidence and variable behavior [1,2]. Gastric carcinoid tumors are an infrequent subtype that represents less than 1% of all stomach neoplasms.

However, their incidence has been increasing, probably due to the growing use of digestive endoscopies [3,4,5]. Type 1 gastric carcinoids (GC-1) are the most frequent subgroup; they are usually associated with chronic atrophic gastritis and hypergastrinemia [6]. Type 2 gastric carcinoid tumors are gastrin-producing tumors (Zolliger–Ellison syndrome) and usually occur in the context of multiple endocrine neoplasia type 1 (MEN-1). Endoscopic findings of Zollinger–Ellison syndrome show normal or hypertrophic mucosa in the context of hypergastrinemia and hyperchloremia, which is a differential finding with respect to GC-1 [7]. Type 3 gastric neuroendocrine tumors occur in isolation, are usually larger in size, and are not related to hypergastrinemia or autoimmune atrophic gastritis [8]. Their behavior tends to be more aggressive than GC-1s [5,8]. In addition, the existence of type 4 gastric neuroendocrine tumors has been suggested, but only isolated clinical cases have been published. GC-4s are usually multiple and have invasive lesions, in the context of hypergastrinemia, in relation to a lack of acid production by hypertrophied parietal cells. These lesions are probably related to an alteration of these cells that prevents hydrochloric acid from being secreted. Pathological studies usually show a different cytology from the type 1, 2, and 3 tumors already reported [9,10,11].

Some of the factors that could explain the continuous increase in the incidence of gastric carcinoids in recent decades could be related to the routine habit of taking biopsies during endoscopic procedures, the development of immunohistochemical techniques, and the deepening knowledge of neuroendocrine oncology. In addition, the implementation of screening in syndromes such as MEN-1 has facilitated the active search and diagnosis of gastro-entero-pancreatic neuroendocrine tumors in these patients [12]. For these reasons, it remains unclear whether this disease is really on the rise or whether it is a reporting artifact [4,13,14]. One of the causes cited for the increased incidence of GC-1 is the widespread use of proton pump inhibitors. The tumorigenesis mechanism is mediated by the effects of gastrin on the CCK2R receptor in ECL cells, which leads to hyperplasia, dysplasia, and finally, neoplasia. However, it has recently been postulated that the degree of response of ECL cells to gastrin is modified by a number of genetic influences, underlying risk factors, and the duration of exposure to hormonal influence. Although the development of GC-1 in regular users of proton pump inhibitors is higher than in the general population, the overall risk for its development is currently considered low [15].

Pathophysiologically, GC-1s are associated with autoimmune atrophic gastritis [3] with the complete atrophy of the oxyntic mucosa, which produces achlorhydria and intrinsic factor deficiency. When achlorhydria persists, the G cells in the gastric antrum experience hyperplasia, and excessive secretion of gastrin occurs [4]. Hypergastrinemia promotes enterochromaffin cell hyperplasia, which may lead to the appearance of GC-1s [14]; these are usually not very aggressive and small in size [8]. Approximately 5% of patients with autoimmune chronic atrophic gastritis will develop a gastric carcinoid tumor, but the finding is usually incidental and the course of the disease is benign [3].

Regarding therapy for GC-1s, although active surveillance can be performed in tumors with diameters less than 1cm, treatment usually requires endoscopic resection [16]. Antrectomy still plays a role when endoscopic resection is not feasible or when poor prognostic factors are present, such as frequent recurrences [17,18,19,20]. However, the use of this technique has been decreasing due to its highly variable long-term results [21].

The inhibitory effects of somatostatin on hormone release and exocrine secretion have been well known for many years. Somatostatin analogues have been shown to be beneficial in the symptomatic treatment of various clinical conditions related to hormonal hyperproduction of GEP-NET, such as carcinoid syndrome, gastrinomas, or VIPomas, as well as their anti-proliferative effects in invasive and/or metastatic GEP-NET [22,23].

The mechanism by which somatostatin inhibits acid secretion is through the somatostatin subtype 2 receptor (SSTR2) of the enterochromaffin cell, which inhibits histamine secretion by interfering with the gastrin-induced calcium signaling [24]. When it binds to this receptor, it inhibits histamine secretion by interfering with the gastrin-induced calcium signaling. In addition, the somatostatin analog octreotide can exert an inhibitory effect on all types of oxyntic cells [25], which has led to its use as a suppressive test to evaluate the outcome of antrectomy in isolated patients with gastric carcinoid tumors. The capability of somatostatin to both directly inhibit enterochromaffin cell proliferation and gastrin release, and thus inhibit its proliferative stimuli, has been known for a long time [24].

Initially, octreotide, administered once daily for 6 months, was shown to reduce fasting gastrin levels and enterochromaffin cells in patients with chronic atrophic gastritis and hypergastrinemia [26]. Subsequently, the direct antiproliferative effects of octreotide on enterochromaffin cells continuously stimulated by gastrin [27] were demonstrated.

This ability of somatostatin analogues to reduce hypergastrinemia related to chronic atrophic gastritis has opened up great therapeutic possibilities for its use in the treatment of GC-1 [18,26].

Over time, case reports have emerged, describing the use of somatostatin analogs in human type 1 gastric carcinoids [28]. Over the last two decades, studies have been carried out that have demonstrated clinical benefits in these tumors [21,29,30,31]. Initially, intermittent treatment with somatostatin analogues was chosen, leading to a decrease in gastrin levels, as well as size decrease or even disappearance of the tumor [32,33,34,35]. However, the efficacy of intermittent treatment was questionable after the withdrawal of somatostatin analogues [36].

The aim of this study was to determine whether somatostatin analogues administered chronically in patients with relapsed GC-1 could have a lasting effect in reducing recurrences in these tumors.

## 2. Material and Methods

In this retrospective, observational study, performed in two centers, patients diagnosed with GC-1 tumors who presented with at least one neoplastic recurrence (confirmed by pathological anatomy) and who were under chronic treatment with somatostatin analogues were selected. Patients with neuroendocrine tumors other than GC-1 and those treated with somatostatin analogues intermittently were excluded. After applying the inclusion and exclusion criteria, among all those eligible, nine patients were enrolled.

Patients were evaluated clinically, biochemically, and endoscopically, in the context of standard clinical practice, prior to initiation of treatment, and thereafter at 6–12 month intervals. Eight patients were treated with Somatulina Autogel (Ipsen, Paris, France), at monthly dosages of 120 mg (n = 5), 90 mg (n = 1), and 60 mg (n = 2). A single patient was treated with octreotide at 30 mg monthly. The median time since diagnosis was 6.4 years and the median time on treatment was 22 months. All patients included in the study were chronically treated with somatostatin analogs. No patient had to discontinue treatment due to adverse effects.

### 2.1. Clinical Evaluation

Subsequent clinical evaluation included anthropometric variables (weight and height), presence of autoimmune atrophic gastritis, other autoimmune diseases, *H. pylori* infection, comorbidities, and standard treatments, including previous use of proton pump inhibitors. GC-1′s treating physicians did not prescribe treatment with proton pump inhibitors. All patients taking proton pump inhibitors were informed of their trophic effect on the gastric mucosa and the difficulties they could cause in follow-up due to the alteration in chromogranin A and gastrin values. Withdrawal of treatment with proton pump inhibitors was proposed to all patients; four of them preferred to continue taking them.

### 2.2. Serum Analysis

Serum gastrin and chromogranin A (CgA) levels were measured prior to initiation of treatment with somatostatin analogues. In patients who were receiving concomitant treatment with proton pump inhibitors, CgA and gastrin values were excluded from the analysis due to the high false positive rate in this setting. Serum gastrin and CgA were measured using commercially available chemiluminescent immunometric asomatostatin analogue kits based on a ligand-labeled mouse monoclonal antibody for gastrin: IMMULITE 2000 XPi Immunoasomatostatin Analogues System (Siemens Healthcare Diagnostics Inc., Deerfield, IL, USA), normal reference range of 40–108 mU/L for gastrin, and KRYPTOR GOLD, (Echevarne Laboratories, Barcelona, Spain; ISO 9001:2000), range of 19.4–98.1 ng/mL for CgA.

### 2.3. Radiological Assessment

Six of the nine patients underwent ^111^In-pentetreotide scintigraphy (Octreoscan) to exclude the presence of distant metastases before the initiation of treatment. Additionally, four patients were subjected to computerized tomography (CT) of the abdomen, and one underwent magnetic resonance imaging (MRI).

### 2.4. Endoscopic Assessment

Upper gastrointestinal endoscopies, with biopsies in the antrum, body, and fundus, were performed periodically in all patients. When possible, visible lesions were removed.

### 2.5. Histopathological Assessment

Sections were immunostained for chromogranin, neuron-specific enolase (NSE), synaptophysin (SYP), and Ki-67 proliferative index using homologous antibody (Dako Omnis). The diagnosis of carcinoid tumors was confirmed morphologically during endoscopy, along with positive immunocytochemical staining for NSE, SYP, and/or chromogranin.

### 2.6. Treatment Response Criteria

Response to treatment was defined following the criteria of the international treatment guidelines as follows: (1) remission (complete regression of all clinical, radiological, and hormonal evidence of the tumor); (2) partial response (50% or greater reduction in all measurable tumors, clinical symptoms, and hormonal levels, with no appearance of new lesions); (3) stable disease (less than 50% reduction or no greater than 25% increase in tumor size, clinical symptoms, and hormonal measurements); (4) recurrence (appearance of new lesions, or a 25% or more increase in tumor size, and clinical/hormonal deterioration) [37,38,39].

### 2.7. Statistical Analysis

Quantitative variables were expressed as median and range, while qualitative variables were presented as counts and percentages. Student’s *t*-test was performed for paired data to analyze two related samples, or their non-parametric variant if the sample distribution was abnormal (Wilcoxon test).

Spearman’s rho analysis was performed to find correlations between changes in clinical–biochemical markers over time (positive rho values indicated a variation in the same direction, while negative rho values indicated a variation in the opposite direction, between two biochemical parameters). A correlation map was made with all the values of Spearman’s rho. Subsequently, a Poisson regression was performed to establish independent predictive factors, adjusted for the recurrence variable. Finally, Cox regression analysis was used to analyze the effect of somatostatin analogues’ treatment on the appearance of tumor recurrences over time, comparing each patient to themselves before starting treatment. For all statistical analyses, the STATA 17.0 BE-Basic Edition package (Lakeway Drive, College Station, TX, USA) was used. A *p* value < 0.05 was considered statistically significant. The study protocol was approved by the local Research Ethics Committee and was carried out according to the Declaration of Helsinki relating to human studies (Study number: 2022-5064).

## 3. Results

### 3.1. Patient Characteristics

Nine patients were evaluated, of which five (55.6%) were female; the median age at the time of diagnosis was 54 years. Six tumors were grade 1 and three were grade 2; none had distant metastases. Median Ki-67 values were 2% (range 1–10). The median size of the primary tumor was 4 mm (range between 2 and 10 mm). The median time from diagnosis to the start of treatment with somatostatin analogues was 47.5 months (range 5.4 to 93.2 months) and the median time with treatment was 22 months (range between 8.1 and 117.2 months).

All the patients were diagnosed with chronic atrophic gastritis, and five of them suffered from other autoimmune diseases. Over the course of the entire follow-up period, one patient died of causes not related to GC-1. The deceased patient was an 89-year-old woman who contracted a COVID-19-related pneumonia. She was the patient with the longest follow-up time with treatment (117 months). Six of the patients (66.6%) presented with multifocal disease. In the remaining three patients, after diagnosis and endoscopic treatment of the first neoplastic recurrence, treatment with somatostatin analogues was offered; all of them accepted. Of these, one patient presented a single tumor recurrence, which was removed endoscopically, after 17 months of treatment with somatostatin analogues. The patient continued with the treatment prescribed chronically after the tumor removal. The total follow-up time for this patient was 22 months. The clinical characteristics of the sample included in the study are shown in Table 1 and Table 2.

Figure 1 displays microscopic images of cases of Type 1 gastric neuroendocrine tumors.

### 3.2. Effect of Somatostatin Analogues Treatment on Biochemical Variables

When studying the effects of chronic treatment with somatostatin analogues, a tendency to decrease in gastrin, from 793 (188–2455) mU/L to 583 (308–1312) mU/L, and Cg A levels from 251 (42–511) ng/mL to 73 (43.9–400) ng/mL, were observed, but the differences were not statistically significant (*p* = 0.34) and (*p* = 0.26), respectively) (Figure 2).

### 3.3. Chronic Treatment with Somatostatin Analogues in the Prevention of Tumor Recurrence

With somatostatin analogues’ treatment, one patient (11.1%) showed complete regression criteria and another developed a tumor recurrence. Seven patients (77.8%) showed a partial response criteria (no clinical and radiological disease, but persistence of gastrin elevation, although this was of little value in four patients that were under treatment with proton pump inhibitors during follow-up). Only one patient experienced tumor recurrence.

Univariate analysis showed a strong and negative correlation between somatostatin analogues’ treatment and tumor recurrences (Spearman’s rho −0.875; *p* < 0.001). A correlation trend was observed between treatment with somatostatin analogues and gastrin levels (Spearman’s rho −0.361; *p* = 0.15). Ki-67 levels did not correlate with recurrences (*p* = 0.91), with tumor size (*p* = 0.88), or with multicentric tumors (*p* = 0.70).

In the multivariate analysis, which adjusted for Ki-67 expression, sex, time of progression, age, and tumor size, chronic treatment with somatostatin analogues was found to be independently associated with a lower number of recurrences (OR 0.054, *p* = 0.005) (Figure 3). This finding suggests a trend towards a protective effect of somatostatin analogues in the reduction in tumor recurrence.

The survival analysis, which used Cox regression to predict tumor recurrence, revealed a trend suggesting that chronic treatment with somatostatin analogues (Figure 4A) provided protection against the onset of new tumor recurrence, when compared to usual clinical practices. This trend remained significant after adjusting for tumor size, gender, age, and Ki-67 expression (Hazard Ratio 0.086 *p* = 0.026; CI 95% 0.001–0.746) (Figure 4B).

## 4. Discussion

Currently, the treatment of choice for GC-1s is endoscopic resection. Endoscopic mucosal resection and submucosal dissection are techniques with a high probability of success and a low rate of complications. For this reason, they are the most commonly used treatment in GC-1 [40,41,42]. However, given the high rate of multifocal tumors and the possibility of submucosal invasion, it is sometimes not possible to perform a complete resection.

The combination of endoscopic resection and endoscopic surveillance appeared to be a very reasonable option in the treatment of GC-1. Some studies achieved complete metastasis-free survival at 4 years follow-up, but with tumor recurrence rates of 60% [43]. In other studies, with larger sample sizes, much lower recurrence rates were observed [16]. This large variation between recurrence rates in the different studies could be due to the difficulty in determining whether the lesions observed at follow-up were true recurrences or simply small lesions missed at the initial endoscopy [21]. Another advantage offered by digestive endoscopy is the monitoring of gastric lesions other than GC-1 in the context of chronic atrophic gastritis. Increased incidence of intestinal-type gastric cancer has been described in up to 23% of patients with GC-1 in long-term follow-up [44]. All this underscores the central role of digestive endoscopy in the diagnosis and treatment of GC-1.

However, there has been a lack of clear instructions in management guidelines and doubts remain about which treatment should be performed. Recently, a 10-year prospective single-center study proposed a size >10 mm as the cutoff diameter level to identify tumors that needed radical resection [45]. According to the current guidelines of ENETS [37], endoscopic management with resection of the lesion represents the gold standard for GC-1s greater than 10 mm that do not infiltrate the muscularis propria and have no evidence of angioinvasion. In GC-1s smaller than 10 mm, annual monitoring is recommended. It has been suggested that surgery should be limited to cases with: clearly demonstrated invasion beyond the submucosa, presence of metastases, multiple tumors, tumor recurrences, and tumors larger than 2 cm in size, although there are no clear procedures in the different published guidelines [37]. Additionally, 68-Gallium PET in GC-1 was recently proposed as a diagnostic test to establish the most appropriate personalized therapeutic strategy, including somatostatin analogues, in selected groups of patients [46].

Although, in the past, somatostatin analogues have not been indicated in the treatment of GC-1 [47], the gradual emergence of studies advocating their use has allowed them to be increasingly used as a therapeutic pillar. The latest guidelines mention its use as a treatment for multiple lesions that are difficult to resect by endoscopy [37]. The action of somatostatin analogues, such as octreotide or lanreotide, which inhibit gastrin secretion and consequent enterochromaffin cell proliferation, provides a mechanism to block the hyperplasia and dysplasia that eventually leads to neoplasia, and could thus prevent the appearance of GC-1. To date, studies with small sample sizes, in which short-term intermittent somatostatin analogue treatment was used, have shown benefits [21,31,32,33,34,48]. In this regard, treatment with intermittent somatostatin analogues achieved complete response rates in up to 76% of patients in a retrospective, multicenter study that included 36 patients with a similar recurrence rate between endoscopic and pharmacological treatment [31]. However, recurrences after discontinuation of intermittent treatment present long-term uncertainties [36]; after withdrawal of the drug, the anti-proliferative effect does not appear to be maintained and the disease often recurs [36]. A study that followed patients treated for 12 months with somatostatin analogues for 5 years found that all patients relapsed after treatment discontinuation and concluded that sustained treatment should be considered in order to achieve lower relapse rates [37]. Similarly, a recent meta-analysis suggested that the duration of therapy was one of the determining factors for successful treatment [29], proposing that a longer duration of treatment with somatostatin analogues in the follow-up of patients with GC-1 could be useful to prevent relapses.

In this sense, our data are consistent with the evidence published in recent years, demonstrating a trend towards reduced tumor recurrence rates with long-term treatment with long-acting somatostatin analogues when compared to active surveillance during medium- and long-term follow-up. This treatment is well-tolerated without side effects requiring discontinuation. Currently, it is the longest follow-up (median 22 months) in a published study of GC-1s receiving chronic treatment with somatostatin analogues. We have provided further evidence for the hypothesis that prolonged treatment without discontinuation with somatostatin analogues shows a trend to a decrease in the risk of recurrence in GC-1 patients and that it could be postulated as a therapeutic alternative to conventional treatment. In our work, treatment with somatostatin analogues was the only related variable, independently of age, years since diagnosis, tumor size, or Ki-67 values, in the prevention of tumor recurrence (Figure 2). Despite that, tumor size seemed to be important in the evolution and management of GC-1 [37,45]. Ki-67 levels had a much smaller impact than in other neuroendocrine neoplasms, such as those of pancreatic or intestinal origin [49,50]. Several studies have examined the influence of Ki-67 on the clinical course of patients with gastric neuroendocrine tumors without establishing clear differences in terms of probability of long-term survival or aggressiveness of the disease [46,51,52]. Regarding the dose and the drugs used, based on the survival analysis performed, treatment with Lanreotide 120 mg monthly as a long-term therapy is recommended, as mentioned in other studies [31]. However, in our work, other molecules and/or doses were used in isolated cases (Table 2).

Regarding other therapeutic options for GC-1, on the one hand, although endoscopy is the technique of choice for the diagnosis and treatment of GC-1, it does not always achieve margin-free resection, regardless of the endoscopic technique used for tumor removal [53]. In addition, the endoscopic procedure is not an easy procedure for some patients. Although all recurrent GC-1s initially require close endoscopic follow-up, the potential decreases in tumor recurrence during somatostatin analogue treatment could allow clinicians to space out the frequency of gastrointestinal endoscopies in selected cases, improving the quality of life of these patients. On the other hand, long-term treatment with somatostatin analogues offers some advantages over surgery. First, most tumors are grade 1. They do not present distant disease [29,30] and some even disappear over time [54], supporting a more conservative approach. There are also no clear surgical indications for total or subtotal antrectomy. This has probably led to an excess of surgical procedures in the past [30]. The long-term benefits of antrectomy remain uncertain [17,55], as recurrence in up to 20% of all cases has been reported, and the surgical approach is more invasive with a higher risk of complications [21]. Additionally, the success rates of the different surgical and medical therapeutic options are very similar; in the two largest studies published to date, the complete response among patients operated and treated with endoscopy was not superior to that of those treated with somatostatin analogues [30,31].

This work had some limitations. First, the retrospective observational design did not allow the elaboration of a thesis of causality; it only provided hypotheses that must be verified in randomized clinical trials. Second, because of the low incidence of GC-1, it is difficult to bring together even individuals from two centers. Our sample size of only nine patients is too small to determine the significance of the results, which can only be considered as a trend observed after applying the proposed therapy. Thus, further studies with larger sample sizes are needed to confirm these preliminary findings. Third, despite increasing scientific evidence, treatment with somatostatin analogues is not a conventional therapy fully implemented in clinical practice. It is necessary to prescribe off-label treatment in selected patients after explaining the risks and benefits. For this reason, some patients prefer to start treatment after the first tumor recurrence, while others prefer to wait longer. Therefore, the follow-up times under treatment were highly variable in our cases, making the interpretation of the results more difficult. Lastly, somatostatin analogues are expensive drugs that, when administered long-term in people expected to survive for a long time, can generate high costs to the system. In our work, a pharmaco–economic study to assess whether the application of chronic treatment with somatostatin analogues in patients with relapsed GC-1 was cost-effective was not carried out.

## 5. Conclusions

Treatment with somatostatin analogues is a feasible option for recurrent GC-1s that are difficult to manage by endoscopy and/or gastrectomy. The medium-term results, as well as the good safety profile, allow us to consider somatostatin analogues therapy as an alternative chronic treatment for selected patients. Further prospective studies and randomized clinical trials will be needed to assess the risk/benefit of sustained treatment with somatostatin analogues in patients at increased risk of recurrence.

## Figures and Tables

**Figure 1 biomedicines-11-00872-f001:**
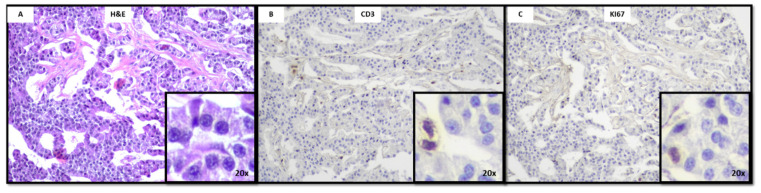
Microscopic images of GC-1 cases. Histological sections of a well-differentiated Type 1 gastric neuroendocrine tumor (**A**) ECL cell tumor-hyperplasia is associated with atrophic gastritis, showing more solid aggregates of cells with moderately hyperchromatic nuclei H&E stain (×10, ×20). (**B**) T-lymphocyte infiltrate using CD3 staining (×10, ×20). (**C**) Ki-67 nuclear proliferative index <2% (Phosphohistone H3 Immunohistochemical stain) (×10, ×20).

**Figure 2 biomedicines-11-00872-f002:**
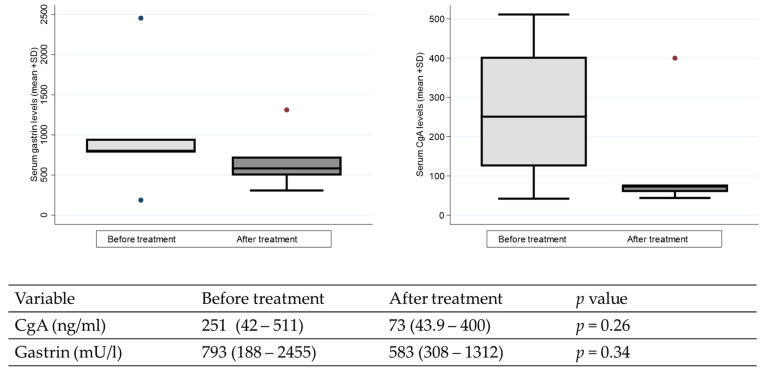
Gastrin and chromogranin A levels before and after treatment with somatostatin analogues. CgA: Cromogranin A.

**Figure 3 biomedicines-11-00872-f003:**
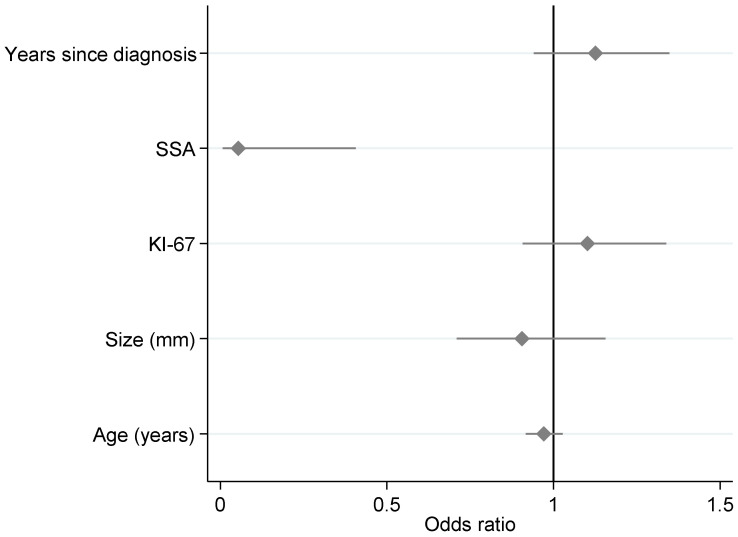
Treatment with somatostatin analogues is shown to be the only independent factor in preventing recurrence of GC-1. (Odds Ratio 0.054; *p* = 0.005). SSA: Somatostatin analogues.

**Figure 4 biomedicines-11-00872-f004:**
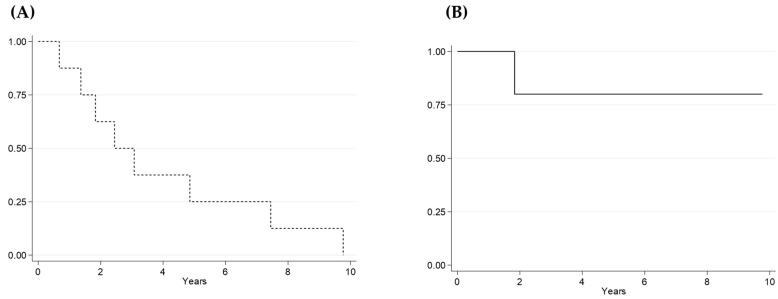
Disease-free survival in 9 patients treated with somatostatin analogues, according to the period of the therapy: without treatment (**A**), and with somatostatin analogues (**B**).

**Table 1 biomedicines-11-00872-t001:** General features of the patients included in the study.

Variable	Obs *n = 9*
Age	54.6 (33.9–89.7)
Sex (Women)	5 (55.6%)
Median time before treatment (months)	47.5 (5.4–93.2)
Median time with treatment (months)	22.0 (8.1–117.2)
Median time since diagnosis (years)	6.4 (1.8–14.5)
Median recurrence before treatment	2 (1–4)
Median recurrence after treatment	0 (0–1)
Ki-67	2 (1–10)
Primary Tumor Size (mm)	4 (2–10)
Chronic atrophic gastritis (%)	9 (100%)
*H. pylori*	1 (11.1%)

**Table 2 biomedicines-11-00872-t002:** Clinical and biochemical characteristics of the study patients.

Nr	Age (Years)	Sex	CAG	CgA (ng/mL)	Gastrin (mU/L)	Tumor Size (mm)	Ki-67 (%)	Multiple	SSA	Pre-Treatment Recurrences	Treatment Recurrences	Time with Treatment (Months)
1	89	W	+	NA	907	2	2	Yes	Lanreotide 60 mg	3	0	117
2	62	W	+	511	953	3	1	No	Lanreotide 90 mg	1	1	22
3	53	W	+	206	789	10	10	Yes	Lanreotide 120 mg	3	0	16
4	47	M	+	296	793	3	3	Yes	Octreotide 30 mg	4	0	89
5	34	M	+	209	1278	5	10	Yes	Lanreotide 120 mg	2	0	29
6	43	M	+	42	188	2	7	No	Lanreotide 60 mg	1	0	8
7	65	W	+	NA	2455	10	2	No	Lanreotide 120 mg	1	0	37
8	55	W	+	27	1629	6	1	Yes	Lanreotide 120 mg	3	0	11
9	64	M	+	15	896	2	1	Yes	Lanreotide 120 mg	1	0	21

W: Woman; M: Man; CAG: chronic atrophic gastritis; NA: Not available; SSA: Somatostatin analogues.

## Data Availability

Data available upon reasonable request.

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
