# Peer review of "Chronic Treatment with Somatostatin Analogues in Recurrent Type 1 Gastric Neuroendocrine Tumors"

_biomedicines, 2023, doi:10.3390/biomedicines11030872_

Round 1

Reviewer 1 Report

The general idea of the study may be good, but the patient sample size is too small to produce statistically relevant results and lot of the statements made by the authors have been well known for a long time. The study might have been of a better value if there was a control group that could confirm that long term treatment with somatostatin analogues is a better approach of small gastric NETs type I than the current standard approach, of endoscopic resection followed by watchful waiting, as their benign behaviour is well known. Here are some comments on the paper:

1.    There are two short forms used for somatostatin analogues, SSA and ASS, please resume to just one.

2.    The authors specify that the grade of all GNETs was G1, however 3 patients had tumours with ki67 index over 3%, which fit them into the G2 category. Please correct this error accordingly.

3.    The authors state that during the follow-up period one of the nine patients died because of COVID19. Was their follow-up period long enough for them to be included in the study? I have also noticed the heterogeneity of the patients’ treatment and its duration (from 8 to 117 months) and I believe that this affects the results of the research, as the follow-up period is very diverse and not fixed for all patients.

4.    AG in Table 2 needs to be explained at abbreviations.

5.    In the Discussion section, the authors state that based on the survival analysis performed, treatment with lanreotide 120 mg or octreotide 30 mg monthly is recommended. However, in section 2 (M&M), the authors mention that only one patient was treated with octreotide. I believe this affects the results, as the sample size of patients treated with octreotide is non-existent and cannot produce these kinds of statements.

6.    I believe that the reasons for the treatment cessation or for the big heterogeneity of the follow-up period are not specified. Were there any side effects of the treatment? Were the shorter follow-up periods due to the late diagnosis of the tumor?

7.    The limitations of the study should be specified at the end of the discussion section, as there are many, the most important being the extremely small patient sample.

8.    An extensive English language check-up should also be performed, as there are some syntax errors that need to be corrected (such as “..being type I gastric carcinoids (GC-1) the majority subgroup…” in the introduction etc.)

Author Response

Referee 1:

Dear Referee 1,

Thank you very much for all your comments.

The general idea of the study may be good, but the patient sample size is too small to produce statistically relevant results and lot of the statements made by the authors have been well known for a long time. The study might have been of a better value if there was a control group that could confirm that long term treatment with somatostatin analogues is a better approach of small gastric NETs type I than the current standard approach, of endoscopic resection followed by watchful waiting, as their benign behaviour is well known. Here are some comments on the paper:

  1. There are two short forms used for somatostatin analogues, SSA and ASS, please resume to just one.

We have revised the entire manuscript and corrected the errors.

  1. The authors specify that the grade of all GNETs was G1, however 3 patients had tumours with ki67 index over 3%, which fit them into the G2 category. Please correct this error accordingly.

There was indeed an error in the transcription of the  tumor grades. We have corrected this error.

  1. The authors state that during the follow-up period one of the nine patients died because of COVID19. Was their follow-up period long enough for them to be included in the study? I have also noticed the heterogeneity of the patients’ treatment and its duration (from 8 to 117 months) and I believe that this affects the results of the research, as the follow-up period is very diverse and not fixed for all patients.

The patient who died from COVID-19 is patient number 1. Her treatment period was the longest of the entire sample (117 months). As you mentioned, the variability in the duration of treatment is a limitation of this study and this limitation has been explained in the Discussion section. As stated, all patients included in this study were treated continuously without drug discontinuation.

  1. AG in Table 2 needs to be explained at abbreviations.

AG means atrophic gastritis. For clarity, we have changed the term to chronic atrophic gastritis

  1. In the Discussion section, the authors state that based on the survival analysis performed, treatment with lanreotide 120 mg or octreotide 30 mg monthly is recommended. However, in section 2 (M&M), the authors mention that only one patient was treated with octreotide. I believe this affects the results, as the sample size of patients treated with octreotide is non-existent and cannot produce these kinds of statements.

We agree with the referee and since octreotide treatment was exceptional in this study, we have removed the explicit mention from the discussion and left only Lanreotide 120 mg monthly, which was usted in the majority of cases.

  1. I believe that the reasons for the treatment cessation or for the big heterogeneity of the follow-up period are not specified. Were there any side effects of the treatment? Were the shorter follow-up periods due to the late diagnosis of the tumor?

All patients in the sample received chronic treatment with somatostatin analogs. No patient had to discontinue treatment due to adverse effects. The shorter follow-up periods are due to the more recent initiation of treatment in these patients.

  1. The limitations of the study should be specified at the end of the discussion section, as there are many, the most important being the extremely small patient sample.

Since the limitations need to be more not sufficiently explicit, we wrote a new paragraph describing them adequately.

  1. An extensive English language check-up should also be performed, as there are some syntax errors that need to be corrected (such as “..being type I gastric carcinoids (GC-1) the majority subgroup…” in the introduction etc.)

Following your suggestion, the manuscript has been professionally proofread in English by MDPI. It has been checked for correct use of grammar and technical terms, and has been edited to a level appropriate for reporting research in an academic journal.

Reviewer 2 Report

This is a manuscript by Fernando Sebastian-Valles et al. entitled “Chronic Treatment With Somatostatin Analogues In Recurrent Type-1 Gastric Neuroendocrine Tumors”. The authors present their results on long term SSA treatment of 9 patients with gastric neuroendocrine tumors type 1.

I have the following comments:

1)    The authors mention at the results section the patients’ characteristics which I suggest that will be moved to the Patients/Material and methods section. At the same paragraph the authors mention that the tumors were all of grade 1 but at the tables 1 and 2 they also include grade 2 tumors with Ki67 of 7%,10% etc.

2)    At the abstract the authors present that “A non-significant tendency to decrease CgA and gastrin was observed due to con-comitant treatment with proton pump inhibitors” Why these patients were treated with PPIs when it si very well known that they have hypochlorydria and PPIs can only augment the trophic effect of gastrin to the ECLcells/gastric neuroendocrine tumors?

3)    Why the patient 2 with a solitary 3 mm large lesion with Ki67 was not followed up and was treated with SSA? The author mention that treatment with SSAs is suggested for recurrent lesions that are difficult to manage by endoscopy or antrectomy.

4)    The patient 6 had a solitary diminutive lesion (2 mm) of a Ki67 of 7%, why it was not removed endoscopically? Similarly, the patient 7 with a solitary lesion of 10mm and Ki67of 2% why it was not removed endoscopically? How the authors justify SSA treatment?

5)    The authors have to state clearly that the patients have to be treated life long according to the data we have up to now. This fact makes SSA treatment an expensive treatment.

6)    The authors state at the discussion that “Endoscopy presents some disadvantages in the treatment of GC-1s, such as technical difficulty of management and a decrease in quality of life due to frequent endoscopic and therapeutic tests,….”. I think that this is misleading because the patients treated with SSAs will anyway be followed up with endoscopy incase they will be lesions autonomous to SSA treatment and I do not think that lesions of some mm or up to 2 cm are difficult to be managed by experienced endoscopists in tertiary centers that also treat NETs. I suggest that the authors will rephrase.

7)    It is not clear to me how these patients were chosen to be treated with SSA. It would be interested if the authors will present clear criteria of inclusion to the study.

8)    I wonder if the authors could present more clearly about what is new at the present manuscript compared to previous studies.

Author Response

Referee 2:

Dear Referee 2,
Thank you very much for all your comments.

This is a manuscript by Fernando Sebastian-Valles et al. entitled “Chronic Treatment With Somatostatin Analogues In Recurrent Type-1 Gastric Neuroendocrine Tumors”. The authors present their results on long term SSA treatment of 9 patients with gastric neuroendocrine tumors type 1.

I have the following comments:

1)    The authors mention at the results section the patients’ characteristics which I suggest that will be moved to the Patients/Material and methods section. At the same paragraph the authors mention that the tumors were all of grade 1 but at the tables 1 and 2 they also include grade 2 tumors with Ki67 of 7%,10% etc.

There was indeed an error in the transcription of the grade of the tumors. We have corrected the error.

2)    At the abstract the authors present that “A non-significant tendency to decrease CgA and gastrin was observed due to con-comitant treatment with proton pump inhibitors” Why these patients were treated with PPIs when it si very well known that they have hypochlorydria and PPIs can only augment the trophic effect of gastrin to the ECLcells/gastric neuroendocrine tumors?

 GC-1's treating physicians did not prescribe treatment with proton pump inhibitors. All patients taking proton pump inhibitors were informed of their trophic effect on the gastric mucosa and the difficulties they could cause in follow-up due to the al-teration of chromogranin A and gastrin values. Withdrawal of treatment with pro-ton pump inhibitors was proposed to all patients; four of them preferred to continue taking them.

3)    Why the patient 2 with a solitary 3 mm large lesion with Ki67 was not followed up and was treated with SSA? The author mention that treatment with SSAs is suggested for recurrent lesions that are difficult to manage by endoscopy or antrectomy.

In patient no. 2, a first lesion was resected in 2016 by endoscopy (the study initiated because of diarrhoea, one polyp was resected, but two other polyps remained). In a revision gastroscopy in 2017 two lesions were resected, neither of them GC-1. New resection of 3 mm GC-1 grade 1 was performed at a new gastroscopy in 2019  and this was considered a pre-treatment recurrence. Treatment with somatostatin analogues was started in 2020. He developed a new GC-1 in a gastroscopy performed at 2022 which was considered  a recurrence despite chronic treatment with somatostatin analogues.

4)    The patient 6 had a solitary diminutive lesion (2 mm) of a Ki67 of 7%, why it was not removed endoscopically? Similarly, the patient 7 with a solitary lesion of 10mm and Ki67of 2% why it was not removed endoscopically? How the authors justify SSA treatment?

Endoscopic resection was performed in both patients when lesions compatible with recurrence were observed on endoscopy. Treatment with somatostatin analogues was started on a chronic basis after the lesions were found to be GC-1 recurrences.

5)    The authors have to state clearly that the patients have to be treated life long according to the data we have up to now. This fact makes SSA treatment an expensive treatment.

We have added in the new paragraph of limitations the fact that chronic treatment with somatostatin analogues is expensive in the long term.

6)    The authors state at the discussion that “Endoscopy presents some disadvantages in the treatment of GC-1s, such as technical difficulty of management and a decrease in quality of life due to frequent endoscopic and therapeutic tests,…”. I think that this is misleading because the patients treated with SSAs will anyway be followed up with endoscopy incase they will be lesions autonomous to SSA treatment and I do not think that lesions of some mm or up to 2 cm are difficult to be managed by experienced endoscopists in tertiary centers that also treat NETs. I suggest that the authors will rephrase.

At the start of follow-up, it is true that patients who have relapsed will continue to have regular endoscopies for short periods of time. However, in cases where the frequency of recurrences decreases with somatostatin analogue treatment, the frequency of endoscopies can be spaced, thus improving patients' quality of life, without the need for testing 2-3 times a year. However, we have rephrased the sentence because it seemed confusing and we have added a paragraph at the beginning of the discussion indicating that endoscopic treatment is the therapy of choice.

7)    It is not clear to me how these patients were chosen to be treated with SSA. It would be interested if the authors will present clear criteria of inclusion to the study.

Inclusion criteria have been added to the new manuscript. They were patients with GC-1 tumours who have relapsed and who are chronically treated with somatostatin analogues. Each patient with these characteristics was offered initiation of treatment after signing the informed consent for off-label drugs. Those who met the above criteria were included in this study.

8)    I wonder if the authors could present more clearly about what is new at the present manuscript compared to previous studies.

Given that the available evidence is in favor of increasingly longer treatment times with somatostatin analogues in patients with GC-1 (Jianu CS et al, Campana et al, Rossi et al), our work provides a cohort from two different centres in which chronic without interruptions was considered.

Reviewer 3 Report

Neuroendocrine gastric tumors are a continually increasing pathology with a significant risk of recurrence, with a therapeutic option of somatostatin analogues. However, currently we have few studies on treatment of recurrence.

The introduction can be improved. Although it generally covers the given topic, significant data is missing. So, I recommend the authors to elaborate a little on the types of gastric carcinoid, on some pathophysiological elements and on the current therapeutic options, including endoscopic treatment criteria, as according to the current practice guidelines. Although it is not a review article, the inclusion of such data sets the basis for the current study, gives fluency to the article and adds value.

The methodology is missing exclusion criteria. The statistics are well-designed.

The references cover essential works on the subtopic and are up-to-date.

In conclusion, it is a well written article on an interesting topic concerning carcinoid tumors. Given the paucity of literature on this topic, as well as the small number of patients who are usually included in these studies, the contribution of the research team can only be welcome in the hope of obtaining as many results as possible that will translate into future management guidelines.

Author Response

Referee 3:

Neuroendocrine gastric tumors are a continually increasing pathology with a significant risk of recurrence, with a therapeutic option of somatostatin analogues. However, currently we have few studies on treatment of recurrence.

The introduction can be improved. Although it generally covers the given topic, significant data is missing. So, I recommend the authors to elaborate a little on the types of gastric carcinoid, on some pathophysiological elements and on the current therapeutic options, including endoscopic treatment criteria, as according to the current practice guidelines. Although it is not a review article, the inclusion of such data sets the basis for the current study, gives fluency to the article and adds value.

The methodology is missing exclusion criteria. The statistics are well-designed.

The references cover essential works on the subtopic and are up-to-date.

In conclusion, it is a well written article on an interesting topic concerning carcinoid tumors. Given the paucity of literature on this topic, as well as the small number of patients who are usually included in these studies, the contribution of the research team can only be welcome in the hope of obtaining as many results as possible that will translate into future management guidelines.

Following your suggestion we have added additional information on pathophysiology, current therapeutic options and endoscopic criteria at the introduction section. In addition, we have drafted inclusion and exclusion criteria to clarify the methodology of the work. Thank you very much for your comments. 

Reviewer 4 Report

Dear Author, 

With all the limitations mentioned in part, such as the small number of patients included in the study, the article brings useful information related to the treatment with SSA for recurrent GC-1s.

I consider the article suitable for publication.

Kind regards,

Author Response

Referee 4:

With all the limitations mentioned in part, such as the small number of patients included in the study, the article brings useful information related to the treatment with SSA for recurrent GC-1s.

I consider the article suitable for publication.

Kind regards,

Thank you very much for your comments.

Round 2

Reviewer 1 Report

Thank you for your prompt answer to my suggestions. The authors have performed the required changes accordingly, improving the scientific soundness of the article. I still believe that the patient cohort is too small to produce statistically relevant results and most of the statements the authors made have been demonstrated before, but, if the Academic Editor agrees with this matter, the paper can be published in the current form.

Author Response

Thank you very much for all your comments.